# Combining Computed Tomography and Histology Leads to an Evolutionary Concept of Hepatic Alveolar Echinococcosis

**DOI:** 10.3390/pathogens9080634

**Published:** 2020-08-04

**Authors:** Johannes Grimm, Annika Beck, Juliane Nell, Julian Schmidberger, Andreas Hillenbrand, Ambros J. Beer, Balázs Dezsényi, Rong Shi, Meinrad Beer, Peter Kern, Doris Henne-Bruns, Wolfgang Kratzer, Peter Moller, Thomas FE Barth, Beate Gruener, Tilmann Graeter

**Affiliations:** 1Institute of Pathology, University Hospital of Ulm, 89081 Ulm, Germany; johannes.grimm@uni-ulm.de (J.G.); annika.beck@uniklinik-ulm.de (A.B.); julianenell@yahoo.de (J.N.); peter.moeller@uniklinik-ulm.de (P.M.); 2Department of Internal Medicine, University Hospital of Ulm, 89081 Ulm, Germany; julian.schmidberger@uniklinik-ulm.de (J.S.); wolfgang.kratzer@uniklinik-ulm.de (W.K.); 3Department of General, Visceral, and Transplantation Surgery, University Hospital of Ulm, 89081 Ulm, Germany; andreas.hillenbrand@uniklinik-ulm.de (A.H.); doris.henne-bruns@uniklinik-ulm.de (D.H.-B.); 4Department of Nuclear Medicine, University Hospital of Ulm, 89081 Ulm, Germany; ambros.beer@uniklinik-ulm.de; 5Department of Infectiology, Central Hospital of Southern Pest National Institute of Hematology and Infectious Diseases, 1097 Budapest, Hungary; dezsenyibalazs13@gmail.com; 6Department of Diagnostic and Interventional Radiology, University Hospital of Ulm, 89081 Ulm, Germany; rong.shi@uniklinik-ulm.de (R.S.); meinrad.beer@uniklinik-ulm.de (M.B.); tilmann.graeter@uniklinik-ulm.de (T.G.); 7Division of Infectious Diseases, University Hospital and Medical Centre of Ulm, 89081 Ulm, Germany; peter.kern@uniklinik-ulm.de (P.K.); beate.gruener@uniklinik-ulm.de (B.G.)

**Keywords:** *Echinococcosis multilocularis*, alveolar echinococcosis, computed tomography, histology, humans

## Abstract

Alveolar echinococcosis (AE) is caused by the intermediate stage of *Echinococcus multilocularis*. We aimed to correlate computed tomography (CT) data with histology to identify distinct characteristics for different lesion types. We classified 45 samples into five types with the *Echinococcus multilocularis* Ulm Classification for Computed Tomography (EMUC-CT). The various CT lesions exhibited significantly different histological parameters, which led us to propose a progression model. The initial lesion fit the CT type IV classification, which comprises a single necrotic area with the central located laminated layer, a larger distance between laminated layer and border zone, a small fibrotic peripheral zone, and few small particles of *Echinococcus multilocularis* (spems). Lesions could progress through CT types I, II, and III, characterized by shorter distances between laminated layer and border zone, more spems inside and surrounding the lesion, and a pronounced fibrotic rim (mostly in type III). Alternatively, lesions could converge to a highly calcified, regressive state (type V). Our results suggest that the CT types mark sequential stages of the infection, which progress over time. These distinct histological patterns advance the understanding of interactions between AE and human host; moreover, they might become prognostically and therapeutically relevant.

## 1. Introduction

Infectious diseases are characterised by a multifactorial interplay between the host and the infective agent. One example is alveolar echinococcosis (AE) [1]. This disease is caused by the larvae of the tapeworm, *Echinococcus multilocularis* (*E*. *multilocularis*), and when untreated, this zoonosis has a poor prognosis [2]. Endemic areas include southern Germany, eastern France, Switzerland, and western China [3]. The life cycle of these parasites begins with the adult worm, which resides in the intestines of canids, such as the red fox. The adult lays eggs, which enter the faeces of the canid host, are then released into the environment. Rodents are intermediate hosts; they ingest the eggs, which then evolve to the larval state of the parasite [4]. Humans can be accidentally infected with the eggs, which leads to AE. AE manifests as an infiltrative tumour-like growth, which forms mostly in the liver. The diagnosis is based on imaging, serology, and histology [2]. The first-line therapy, recommended by the World Health Organisation (WHO) Informal Working Group on Echinococcosis, is a complete surgical resection, accompanied by treatment with benzimidazole derivatives [5]. The WHO-PNM classification focuses on the extent of AE (i.e., Parasitic mass in the liver, involvement of Neighbouring organs, Metastasis) [6].

Morphology of AE liver lesions can be assessed with different classification systems, based on the various imaging modalities. For example, the *Echinococcus multilocularis* Ulm Classification (EMUC) is used to assess ultrasound images (EMUC-US) [7] and computed tomography images (EMUC-CT) [8]. Kodama et al. proposed an MRI classification for assessing MRIs [9]. The EMUC-CT has categorised five distinct morphologies (Types I–V), which can be identified on CT images (Appendix A) [8].

Histological analyses are guided by different structures identified in AE lesions [10]. The most important diagnostic parameter is the detection of the glycoprotein-rich, enveloping layer of the parasite, i.e., the laminated layer. Small particles of *Echinococcus multilocularis* (spems) have been described in immunohistochemical staining with the monoclonal antibody Em2G11 (mAbEm2G11) [11]. This antibody is specific to the Em2 antigen expressed by *E. multilocularis* larvae [12]. Spems are most likely produced when the laminated layer is shed during metacestode growth. Spems can be found in the necrotic area of a lesion, in the adjacent liver tissue, and even in lymph nodes [11,13,14].

In the present study, we compared the various primary morphological types defined with the EMUC-CT to corresponding histological findings in liver AE lesions. We aimed to gain insight into a possible time sequence characterised by initial and/or more advanced stages of the disease.

## 2. Materials and Methods

### 2.1. Study Design

Based on pre-therapy CTs, two experienced radiologists (TG, RS) used the EMUC-CT to classify 68 lesions in 63 patients with histologically confirmed AE in the liver (Appendix A) [8]. The classification is based on two pillars: the primary morphology type and the pattern of calcification (Appendix A). In this study, we focused on the main pillar of classification, the primary morphology, because it describes the lesion morphology in CT images. The calcification pattern was not included because it does not describe the shape of the lesion. We had to exclude 19 patients due to the poor quality of the histological specimens. Of the remaining 44 patients, we evaluated 45 lesions, based on 80 representative tissue blocks acquired from partial hepatectomies. All histological analyses were performed with formalin-fixed, paraffin-embedded tissues (Appendix A). The 45 lesion samples were pseudonymised for blinded analyses. Samples were examined to identify different histological parameters by three experienced pathologists (JG, AB, TFEB) with a multihead microscope. At least ten lesions per type were included for CT types I–IV. For this analysis, type III was not divided into types IIIa and IIIb, because these types were only distinguished by lesion size. Due to the rareness of CT type V lesions, we examined only two type V lesions. Due to the small number in this group, these samples were not included in the comparative analyses. In addition, one patient had two lesions of two different CT types. Each of these lesions was evaluated as a single sample. PNM for the actual collective is indicated in Appendix A. Serology data were available (Appendix A) but excluded from further evaluations due to different time points of resection/biopsy and determination.

This study was approved by the Ethics Committee of the University Ulm. It was conducted in accordance with the Declaration of Helsinki (ref. No. 440/15 and 116/13). All study participants provided written informed consent to an anonymised analysis prior to study enrolment.

### 2.2. Staining

We performed haematoxylin and eosin (HE) staining, Periodic Schiff staining (PAS), and Azan staining, according to standard protocols [15,16].

For the immunohistochemical analysis, we used the primary antibody, mAbEm2G11 (IgG_1_), which specifically targeted the *E*. *multilocularis* antigen, Em2 [11,12]

### 2.3. Histological Parameters

The histological parameters that characterised the larval state of *E*. *multilocularis* in the human liver were defined as follows (concentric zones were numbered from the central lesion to the periphery):Zone 1: all lesions had a central necrotic area of various diameters, intermingled with laminated layer fragments of various sizesZone 2: an inner ring next to the necrotic zone, characterised by epithelioid cells and neutrophilic granulocytesZone 3: a fibrotic ring of varying widthsZone 4: an outer ring that contains lymphocytesZone 5: surrounding hepatic tissue

The border zone was defined as zone 2 to zone 4 (Figure 1A).

We analysed tissue sections of liver lesions to determine the histological characteristics of the laminated layer, the border zone, the necrotic area, and the distribution of spems.

Metric measurements were performed with a photo microscope (Axiophot, Oberkochen, Germany), coupled to a Charge-coupled Device (CCD) camera (JVC, KY-F75U, Yokohama, Japan), and Diskus software (Hilgers Technisches Büro, Königswinter, Germany).

#### 2.3.1. Laminated Layer

For the analysis, we measured the thicknesses of laminated layer fragments at ten positions on each PAS-stained slide (Figure 1C). Moreover, we measured the minimum distance between zone 2 and the laminated layer in zone 1 at five different positions (Figure 1A). In addition, we recorded the location of the laminated layer fragments within the necrotic area; the location was recorded as a centralised laminated layer with alveolar-like structures (≤10 alveolar-like structures; magnification 12.5×) or a distributed laminated layer, with tubular laminar fragments distributed throughout the entire necrotic zone (>10 alveolar-like structures; magnification 12.5×; Appendix AA).

#### 2.3.2. Border Zone

The border zone comprises a barrier between the host tissue and the parasite. It includes zones 2 to 4. We measured the widths of all three zones in HE stained sections. The border zone was also measured in nine Azan-stained samples to confirm the HE-stain results. Furthermore, we analysed tissues for the presence/absence of a fibrotic area between the lymphocyte zone and the host liver tissue.

#### 2.3.3. Necrosis

We detected various patterns of necrosis on the tissue sections. We distinguished between a single necrotic area and multiple necrotic areas in a single section.

#### 2.3.4. Small Particles of Echinococcus Multilocularis (Spems)

We analysed the distribution of spems in the necrotic zone. Distributions were scored 1–3 (Appendix A). We also analysed the host liver tissue next to the lesion for the presence/absence of spems. Spems were identified with mAbEm2G11 immunochemistry.

Furthermore, we examined the relative area covered by mAbEm2G11-positive stained spems in a single observation field (magnification 100×). We analysed images of five different observation fields with CellProfiler software (Broad Institute of Harvard and MIT, Cambridge, MA, USA) [17].

### 2.4. Statistical Analysis

Statistical analyses were performed with SAS, Version 9.4 (SAS Institute Inc., Cary, NC, USA). The data were first tested to determine the position and scattering dimensions (mean, standard deviation, median, minimum, and maximum). The Shapiro-Wilk test was used to determine whether data were normally distributed. Pearson’s chi-squared and the exact Fisher test were used to evaluate differences in frequency between two variables. Differences in metric variables measured among the four CT types were evaluated with post hoc Tukey and Kruskal-Wallis tests, based on an analysis of variance (ANOVA). For all tests, two-sided *p*-values < 0.05 were interpreted as statistically significant with a five percent probability of error.

## 3. Results

### 3.1. Patient Cohort

The patient cohort had a mean age of 43.91 years ± 15.10 (range: 20–70), with 12 males (27%) and 32 females (73%). Most patients (*n* = 26, 59%) were 18–30 years; nine patients (20%) were 31–40 years, and nine patients (20%) were > 40 years. There were no statistical differences in the age distribution between men and women (44.25 years ± 15.58 (range: 21–70) vs. 43.77 years ± 15.18 (range: 20–68); *p* > 0.05). The distribution of CT morphology types regarding patient sex is given in Appendix A.

### 3.2. Laminated Layer

We investigated two metric values to evaluate the laminated layers in different CT types. The thickness of the laminated layer was not significantly different among the CT types (*p* = 0.9). However, the distance between the laminated layer and the border zone was a significant discriminating factor among the CT types (*p* = 0.0003). The largest distance was found in type IV lesions (2037.75 ± 968.43 µm), and the shortest distances were found in types I (318.16 ± 266.48 µm) and II (464.26 ± 502.04 µm). Type III lesion distances were intermediate (756.60 ± 969.42 µm). The differences in comparison of type IV to type I (*p* < 0.0001), type II (*p* = 0.0002) and type III (*p* = 0.0069) were significant (Figure 2A).

We also discriminated between samples with centralised laminated layers and those with laminated layers distributed over the whole necrosis (Appendix AA). The frequencies of these distributions were significantly different among the four lesion types (Χ^2^ = 33.44; *p* < 0.0001). In type IV lesions, the laminated layer was localised centrally, with one or very few alveolae in 10/11 (91%) cases. All samples of the other three CT types had distributed laminated layers (types I–III: 100% of samples). The difference between type IV and the other lesion types was significantly different: type I (*p* < 0.0001), type II (*p* < 0.0001), and type III (*p* = 0.0003). Therefore, the type IV lesion was associated with a centralised, alveolar-like laminated layer (*p* < 0.0001) (Table 1 and Table 2).

### 3.3. Border Zone

The width of the entire border zone (zones 2–4) was not significantly different among CT type I-IV lesions. However, when the fibrotic and the lymphocyte-infiltrated zones were analysed, significant differences emerged. The widest fibrosis (zone 3) was found in type III lesions (212.92 ± 113.66 µm), and these differences were significant between the various lesion types (*p* = 0.0089). Types I and II lesions (type I = 147.61 ± 35.27 µm; type II = 147.52 ± 54.46 µm) had intermediate zone 3 widths. The narrowest zone 3 were found in type IV lesions (102.74 ± 37.06 µm; Figure 2B). Among the lesion types, the differences were significant (*p* = 0.0089). In particular, the width of the fibrosis (zone 3) was significantly different between type III and type IV (*p* = 0.0066).

The lymphocyte zone (zone 4) was narrowest in type IV (83.65 ± 33.71 µm) and widest in type III lesions (125.78 ± 32.36 µm). In general, zone 4 widths were significantly different among the four primary morphological types (*p* = 0.0297), particularly between type III and type IV lesions (*p* = 0.0141; Figure 2C).

In some cases, we identified another zone of fibrosis between the zone of lymphocytes and the surrounding normal liver tissue. We defined this histological feature as the outer fibrotic rim (between zones 4 and 5), and we called it zone 3b (Figure 1B). The percentage of samples that exhibited this zone was significantly different among all groups analysed (Χ^2^ = 19.32; *p* = 0.0002). Zone 3b was found in 7/10 (70%) type III samples, in 1/11 (9%) type I samples, in 1/11 (9%) type II samples, and in no type IV samples. Zone 3b was found in type III samples significantly more frequently than in the other three lesion types (type I: *p* = 0.0075; type II: *p* = 0.0075; type IV: *p* = 0.0010; Table 2).

### 3.4. Necrosis

The tissue samples were categorised according to whether a single necrotic area or multiple necrotic areas were detected. The percentages of samples with single and multiple necrotic areas were significantly different among the different lesion types (Χ^2^ = 16.2882; *p* = 0.0010). Type IV lesions had the largest percentage of samples with single necrotic areas (10/11; 91%). Type II lesions had nearly the same number of samples with single and multiple necrotic areas (five multiple necrotic areas, 45%; six single necrotic areas, 55%). The two other lesion types had mostly multiple necrotic areas (type I: 10/11, 92%; type III: 7/10, 70%). The percentages of samples with single and multiple lesions were significantly different between type IV and types I (*p* = 0.003) and III (*p* = 0.0075) lesions (Table 2).

### 3.5. Analysis of Small Particles of Echinococcus Multilocularis (Spems)

Next, we focused on differences in the distribution of spems among the CT types. We analysed spems in the necrotic area surrounding the laminated layer and in the non-necrotic liver tissue outside the border zone (zone 5). In the necrotic area, spem distributions were scored as follows: 1 = spems only attached to the laminated layer; 2 = spems distributed throughout the necrotic area, but weakly stained near the border zone; and 3 = spems distributed throughout the entire necrotic area (Appendix AB). With this approach, spems were absent or weakly stained (scores 1 and 2) more frequently in type IV lesions (8/11; 73%) than in type I (6/11; 55%), type II (7/11; 64%), and type III (5/10; 50%) lesions (Table 2).

We observed a significantly different distribution of spems in the adjacent non-necrotic liver tissue (Χ^2^ = 8.6214; *p* = 0.0348). In these surrounding liver tissues, few type IV samples showed detectable spems (4/11; 36%). Most spems in non-necrotic tissues were observed in type III lesions (9/10; 90%); this finding was statistically significant (*p* = 0.0237; Table 2).

### 3.6. Type V

Due to the small number (*n* = 2) of CT type V cases, we did not include type V in the comparative analyses. The results for type V lesions are shown in Table 1 and Table 2. Calcified areas appeared to be a characteristic feature of type V lesions. Significant values are shown in the heatmap table in Figure 3.

## 4. Discussion

This study was the first to correlate CT findings with histological parameters in AE. We aimed to link histological patterns characteristic of hepatic *E*. *multilocularis* lesions to the different primary morphological EMUC-CT types. We hypothesised that the differences in the histological parameters among the CT types might reflect different kinds of host reactions, and thus, they might provide hints about the time course of the infection. A time-course-based model for *E*. *multilocularis* infections has been described in mice. There, the initial unilocular stage was followed by a multilocular stage during the first week of infection [18,19]. In rodents, the intermediate hosts, the infection progresses more rapidly and aggressively compared to infections in humans, which are considered dead-end hosts [18,19]. Therefore, we hypothesised that the AE infection course in humans might be only partly comparable to the infection course in mice. Indeed, human infections exhibit a much longer disease course (asymptomatic period 5–15 years) due to the more efficient control mechanisms in humans [2].

Based on our histological and imaging data, we defined the CT type IV lesion as the initial stage. This stage was characterised by a small, well-circumscribed lesion with a small number of laminated layers, few spems, little fibrosis, and few inflammatory cells controlling the lesion. Our findings suggested that lesions in the initial stage might progress by extending the laminated layers and spems, as found in CT type I and II lesions. We defined these lesions as progressive stage. Then, the lesions might evolve to an advanced stage that corresponds to CT type III lesions, characterised by pronounced fibrosis, a high burden of spems, and large necrotic areas, which indicate a highly complex, tubular growth pattern.

Alternatively, the initial stage might follow by a dormant stage with no disease progression, or the infection might even regress, due to total necrosis or increasing calcification, as observed in type V lesions (Figure 4) [8]

This proposed AE evolution model includes various stages of disease, based on the various characteristics we observed in the different types of lesions.

The germinal layer represents the vital cellular part of *E*. *multilocularis* in the larval state [20,21]. The most prominent histological structure of *E*. *multilocularis* in the larval state is the laminated layer, which consists of mucins. Mucins protect the germinal layer from the immune cells of the host [22,23]. We found that different lesion types had significantly different distances between the laminated layer and the border zone. This distance was significantly shorter in progressive and advanced stages compared to the initial stage, which had a larger, isolating necrotic area. Furthermore, the localisation of the laminated layer differed in the various lesions. Initial lesions showed only a few central alveolar structures surrounded by necrotic tissue which contain the laminated layer. This morphology was similar to the so-called unilocular morphology in the very early stage of larval manifestation in the mouse [18,19]. In the other lesion types, the distribution of laminated layer fragments was less centralised; thus, this distribution reflected more advanced stages, which were characterised by extended tubular growth.

The first line of interaction between the host and the parasite is the border zone. The border zone includes three distinguishable zones: the inner zone of epithelioid cells and granulocytes (zone 2), the middle zone of fibrotic tissue (zone 3), and an outer layer of B and T lymphocytes (zone 4) [24]. Enhanced fibrosis, occasionally accompanied by an additional, outer fibrotic layer (zone 3b), was a special feature of the advanced stage. Further, the lymphocyte zone (zone 4) was significantly wider compared to the earlier stages. In contrast, initial lesions showed less fibrosis and a narrower lymphocytic rim.

Previous studies have described advanced AE infections accompanied by a strong, scar-like fibrotic reaction [25]. Fibrosis is a well-known mechanism of chronic inflammation. It acts to separate necrotic tissue from vital tissues, and it is generally linked to the time of inflammation; e.g., during chronic liver diseases, hepatitis can lead to cirrhosis [26]. Our data showed fibrotic patterns that supported the evolution model. Moreover, we observed increases in lymphocytes from initial lesions to progressive lesions and from progressive to advanced lesions. This observation supported the idea that the host response to the metacestode was augmented over the course of disease. Past studies showed a conversion of the cellular immune response from an acute inflammatory Th1-dominant response to a Th2-dominant response during the course of infection [23,27].

The presence of one or multiple necrotic areas per histological section analysed could provide hints about the complexity of the lesion. The EMUC-CT [8] defined an initial lesion as a small, spherical focus, which appeared as a single, solid necrotic area in histological analyses. In contrast, the more complex lesions of progressive and advanced stages revealed multiple necrotic areas. Our data suggested that multiple necrotic areas resulted from different views of the same lesion with a tubular growth pattern. This observation was supported by the three-dimensional reconstruction of *E*. *multilocularis* larval growth in human hepatic tissue [28]. In individual cases, it cannot be ruled out that larger lesions may arise from confluent smaller lesions. However, the findings of Tappe et al. further suggest primarily centripetally growing lesions with a complex tubular growth pattern with distinct lesion centres as shown by imaging [28].

Our sequential model of infection was supported by our data on spem distribution. Initial lesions showed significantly fewer spems in the surrounding liver tissue and in the necrotic area. The spem number was highest in the adjacent liver tissue of advanced stage lesions. Spems most likely result from laminated layer shedding over the course of larval growth [11,13]. Thus, our findings suggested that a longer infection duration was accompanied by a higher number of spems, which corresponded to an advanced stage. Previous studies showed that, in *E*. *granulosus*, particles that originated from the laminated layer had an immunomodulating effect, which led to an unconventional maturation of dendritic cells [29]. Furthermore, the Em2 antigen located on spems was identified as a T-cell-independent antigen [30]. Thus, further study is warranted to determine whether spems influence the immune system of the host.

Numbering of types I to V was made only in context with the development of the initial diagnostic/descriptive EMUC-CT in 2016. With the numbering of the types, the descending frequencies of the primary morphologies from type I to V within the collective investigated at that time (*n* = 228) were taken into account.

In view of the current histopathological results, the EMUC-CT classification scheme will be adjusted. The sequential lesion evolution will then be taken into account by a corresponding adapted sequence of the numbered lesion types. Recently published studies regarding epidemiological CT data from Europe and China, as well as the analyses of different CT morphological types concerning their density and their activity in positron emission tomography further support this hypothesis of an evolutionary model in hepatic AE lesions [31,32,33].

The histological features of initial lesions, which correspond to EMUC-CT type IV lesions, could have important implications for routine clinical practice. In specifying an unclarified liver lesion, the gold standard is the biopsy. In an *E*. *multilocularis* infection, detection of the laminated layer is the strongest histological diagnostic criterion. Due to the centralised alveolae in initial lesions, the probability of detecting the laminated layer is low in biopsies acquired from the lesion periphery. Furthermore, few spems are found in the periphery of initial lesion (CT type IV). Therefore, when such a lesion is suspected, based on CT or ultrasound investigations, the biopsy should be acquired from the central part of the lesion, and immunohistochemical staining should be performed with the mAbEm2G11 antibody [11,12]. The various features of initial lesions might directly influence therapy planning, in decisions of whether the lesion might be treated exclusively with pharmacotherapy and whether the therapy duration might be shortened.

Our results suggested that different histological patterns of AE infections could be linked to different disease stages. In the future, these links should be investigated further, because the results might impact clinical decisions regarding prognoses and therapeutic options.

## Figures and Tables

**Figure 1 pathogens-09-00634-f001:**
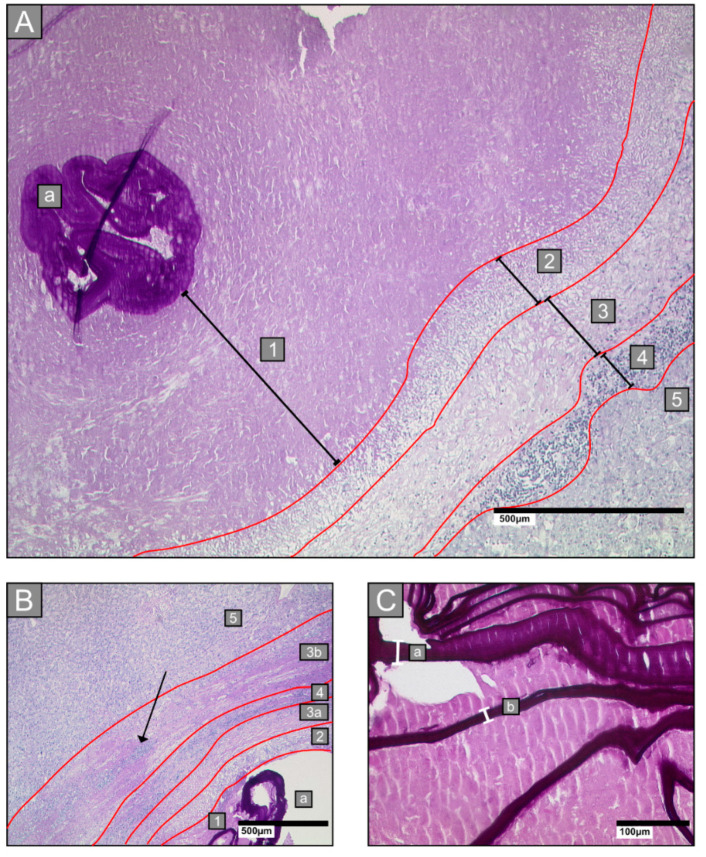
Examples of histological analyses of liver alveolar echinococcosis infections. Sections were stained with Periodic acid–Schiff. (**A**): Infections by *Echinococcus multilocularis* larvae are characterised by five zones: zone 1: necrotic area, surrounding the laminated layer (a); zone 2: inner layer of epithelioid cells and granulocytes; zone 3: fibrotic area; zone 4: outer layer of lymphocytes; zone 5: adjacent liver tissue. (**B**): An example of zone 3b, which is particularly apparent in *Echinococcus multilocularis* Ulm Classification for Computed Tomography (EMUC-CT) type III lesions. This zone comprises an additional fibrotic rim between zone 4 and zone 5. Lymphocyte infiltrates can also be found in zone 3b (arrow) (a: laminated layer; 1: necrotic zone; 2–4: border zone; 5: liver host tissue; 3b: outer fibrotic rim). (**C**): Examples of laminated layer width measurements (indicated with white brackets labelled a and b) in Periodic Schiff staining (PAS)-stained tissue (a = 26.7 µm; b = 14.1 µm).

**Figure 2 pathogens-09-00634-f002:**
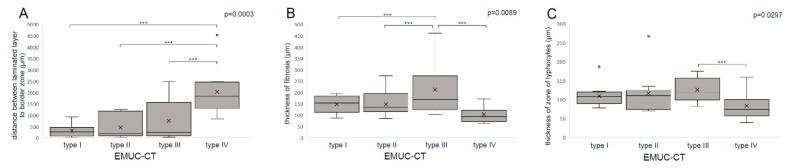
Examples of histological metric measurements. (**A**): Distance between the laminated layer and the border zone. (**B**): Thickness of fibrotic zone 3. (**C**): Thickness of lymphocytic zone 4; *** *p* < 0.001.

**Figure 3 pathogens-09-00634-f003:**
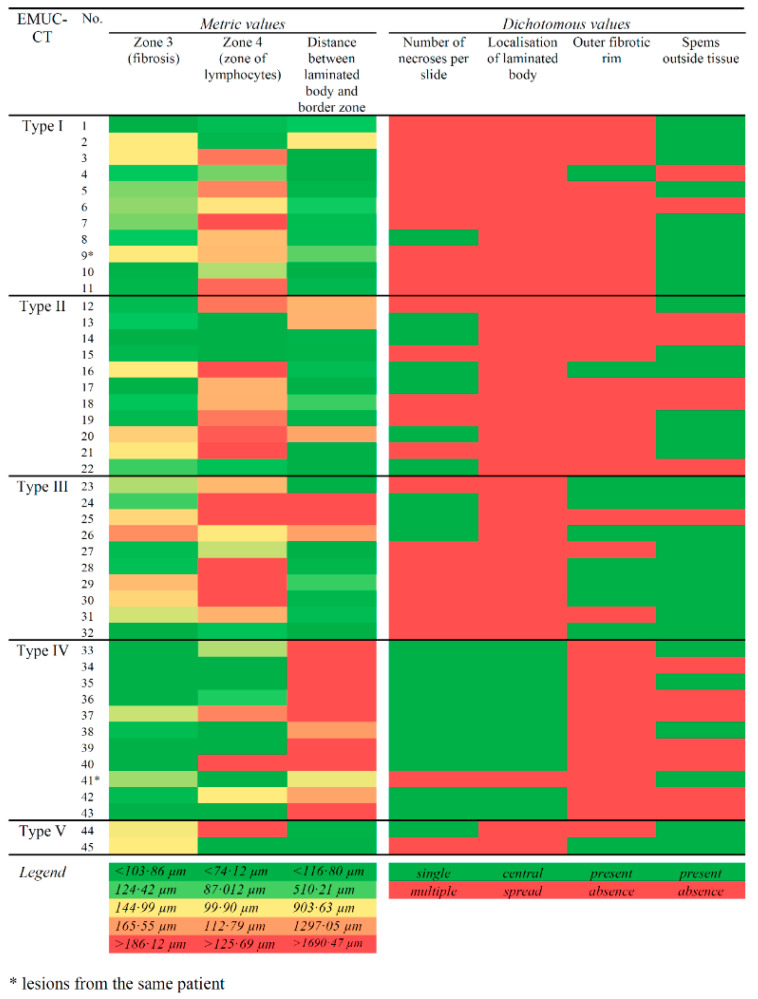
Heatmap of parameters that showed significant results (*p* < 0.05) in the five EMUC-CT types of alveolar echinococcosis lesions. (*Left*) Colours indicate different values of metric values. (*Right*) Dichotomous measures are indicated with red or green. Legend gives the specific definitions for each colour in each column.

**Figure 4 pathogens-09-00634-f004:**
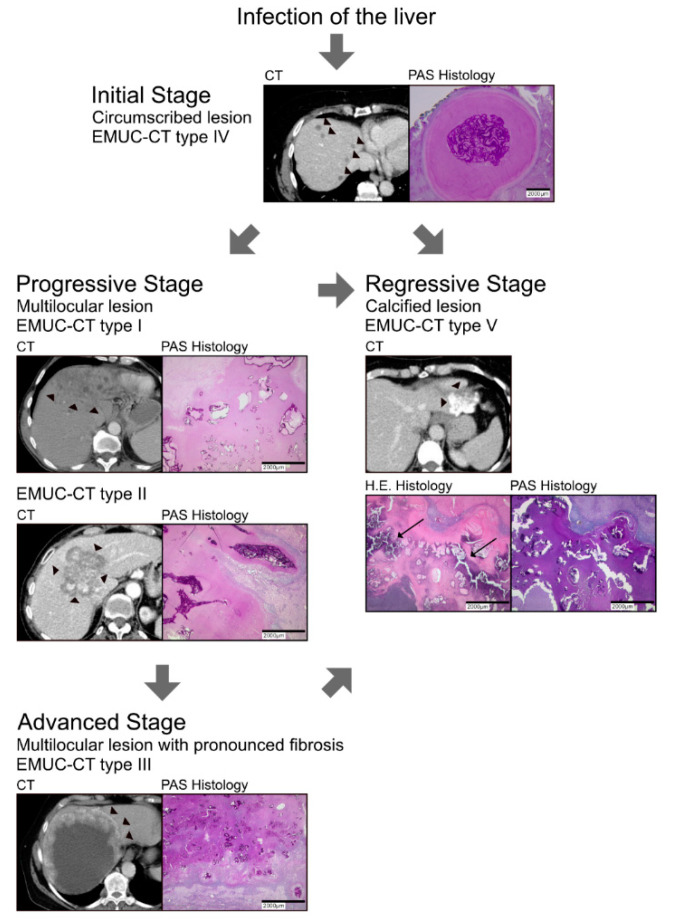
Proposed model of the evolution of alveolar echinococcosis infections in the liver. (*Top*) The initial stage (EMUC-CT type IV) is characterised by a small, well-circumscribed lesion (*left*, arrowheads show multiple small lesions) with few laminated layers (*right*, intense PAS stain). (2nd row) A lesion in the initial stage can progress in two ways: (*left*) it can develop to the progressive stage (CT types I and II) when the laminated layer spreads out (arrowheads show the edge of the lesion); alternatively (*right*), it can eventually become calcified (arrows in HE histology) and regress (regressive stage, CT type V). (3rd row) The progressive stages can evolve to an advanced stage (CT type III; arrowheads show the edges of the lesion).

**Table 1 pathogens-09-00634-t001:** Quantitative histological measurements of alveolar echinococcosis lesions in liver tissues.

Histological Parameter	Lesion Measurements; Mean ± SD Median (Min–Max)					*p*-Value ^b^
	Type I	Type II	Type III	Type IV	Type V ^a^	
Width of border zone (µm)	377.80 ± 64.53 375.99 302.87–487.18	391.83 ± 109.36 361.37 271.41–584.91	454.07 ± 127.49 416.34 286.74–672.44	334.27 ± 103.34 291.47 224.80–573.34	388.54	0.0621
Zone of epithelioid cells (µm)	120.84 ± 26.24 128.02 70.01-165.28	127.76 ± 32.73 116.51 79.74-188.09	115.37 ± 21.98 110.23 90.67-154.85	147.88 ± 77.78 126.20 47.22–340.68	106.33	0.5965
Zone of fibrosis (µm)	147.61 ± 35.27 153.27 87.71–195.12	147.52 ± 54.46 133.59 84.35–272.71	212.92 ± 113.66 169.03 101.90–461.28	102.74 ± 37.06 94.14 62.97–171.22	184.09	0.0089 **
Zone of lymphocytes (µm)	109.36 ± 29.91 107.62 77.37–186.93	116.55 ± 55.23 109.73 69.88–267.44	125.78 ± 32.36 117.76 82.10–174.93	83.65 ± 33.71 74.12 38.96–159.10	98.13	0.0297 *
Thickness of laminated layer fragments (µm)	24.67 ± 7.36 23.89 12.64–37.41	22.85 ± 9.36 24.07 8.86–41.86	46.68 ± 57.17 22.64 15.68–193.32	27.14 ± 12.06 27.95 11.83-46.95	22.72	0.9000
Distance between laminated layer and border zone (µm)	318.16 ± 266.48 264.80 28.00–926.92	464.26 ± 502.04 180.30 30.10–1266.50	756.60 ± 969.42 238.04 38.40–2479.60	2037.75 ± 968.43 1842.40 842.80–4533.80	120.62	0.0003 ***
Spems in necrotic area (relative area covered by mAbEm2G11-positive stained spems in a single observation field; magnification 100×)	0.51 ± 0.19 0.54 0.17–0.81	0.34 ± 0.24 0.23 0.07–0.68	0.38 ± 0.25 0.43 0.04–0.80	0.32 ± 0.20 0.31 0.01–0.66	*-*	0.2621

* *p* < 0.05; ** *p* < 0.01; *** *p* < 0.001. ^a^ Type V was excluded from the analysis due to the small number of lesions. ^b^
*p*-values were calculated only for types I–IV.

**Table 2 pathogens-09-00634-t002:** Semi-quantitative histological measurements of alveolar echinococcosis lesions in liver tissues.

Histological Parameter		Number of Lesions (Percent for Each Type)					*p*-Value ^b^
		Type I	Type II	Type III	Type IV	Type V ^a^	
Localisation of laminated layer fragments	Distributed throughout the necrotic area	11 (100.00)	11 (100.00)	10 (100.00)	1 (9.09)	2 (100.00)	<0.0001 ***
	Centralised alveolae	0 (0.00)	0 (0.00)	0 (0.00)	10 (90.01)	0 (0.00)	
Number of necrotic areas per section	Multiple necrotic areas	10 (90.91)	5 (45.45)	7 (70.00)	1 (9.09)	1 (50.00)	0.0010 **
	Single necrotic area	1 (9.09)	6 (54.55)	3 (30.00)	10 (90.91)	1 (50.00)	
Outer fibrotic rim (Zone 3b)	No fibrosis	10 (90.91)	10 (90.91)	3 (30.00)	11 (100.00)	1 (50.00)	0.0002 ***
	Fibrosis	1 (9.09)	1 (9.09)	7 (70.00)	0 (0.00)	1 (50.00)	
Spems in necrotic area (scored 1–3)	1 (spems only attached to the laminated layer)	0 (0.00)	0 (0.00)	2 (20.00)	2 (18.18)	0 (0.00)	no *p*-value calculated
	2 (spems in entire necrotic area, but less abundant near the border)	6 (54.55)	7 (63.64)	3 (30.00)	6 (54.55)	0 (0.00)	
	3 (spems in entire necrotic area)	5 (45.45)	4 (36.36)	5 (50.00)	3 (27.27)	2 (100.00)	
Spems in non-necrotic liver tissue	No spems	2 (18.18)	5 (45.45)	1 (10.00)	7 (63.64)	0 (0.00)	0.0348 *
	Spems	9 (81.82)	6 (54.55)	9 (90.00)	4 (36.36)	2 (100.00)	

* *p* < 0.05; ** *p* < 0.01; *** *p* < 0.001. ^a^ Type V was excluded from the analysis due to the small *n*. ^b^
*p*-values were calculated only for types I–IV.

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
