# Peer review of "Combining Computed Tomography and Histology Leads to an Evolutionary Concept of Hepatic Alveolar Echinococcosis"

_pathogens, 2020, doi:10.3390/pathogens9080634_

Round 1
Reviewer 1 Report
This study proposed an evolution model for Hepatic Alveolar Echinococcosis by combining Combining Computed Tomography and Histology. The work may have been a laborious one that I should appreciate the authors' time and patience to come up with some results. However, there are several problems that deduct from the quality of this manuscript. Below are several comments on this work.
- Why did not the authors take the type V into consideration?
- I was wondering the data used in this study are collected from Human patients? I have one doubt that a single dataset from one species is enough or not?
- The authors may add more state-of-the-art references to this study to introduce more latest development in this topic.
Reviewer 2 Report
Dear sir,
thank you to select me to review manuscript: Grimm J et al. Combining Computed Tomography and Histology Leads to an Evolutionary Concept of Hepatic Alveolar Echinococcosis. Authors correlated computed tomography scans with histology to identify distinct characteristics for different lesion types. different histological patterns of AE infections could be linked to
different CT scans. Paper is well written, statistical analysis, presentation of results, discussion and conclusions are adequately presented.
Please put part Material and methods before the Results section (at line 75). This error probably occurred during the technical processing of the manuscript. the article can be published after this technical revision.
My final decision is acceptation.
Author Response
Thank you very much for this positive feedback!
Round 2
Reviewer 1 Report
The authors have addressed all my comments.